# CONTROLLED DENOISING FOR DIFFUSION MODELS

## ABSTRACT

Aligning diffusion models for downstream tasks often requires finetuning new models or costly inference-time solutions (e.g., gradient-based guidance) to allow sampling from the reward-tilted posterior. In this work, we explore a simple and low-cost inference-time gradient-free guidance approach, called conditional controlled denoising (`C-CoDe`), that circumvents the need for differentiable guidance functions and model finetuning. `C-CoDe` is a block-wise sampling method with adjustable conditioning on a reference image applied during intermediate denoising steps, allowing for efficient alignment with downstream rewards. Experiments demonstrate that, despite its simplicity, `C-CoDe` offers a balanced trade-off between reward alignment, prompt instruction following, and inference cost, outperforming state-of-the-art baselines. Our code is available at: https://anonymous.4open.science/r/CoDe-Repo.

## 1 INTRODUCTION

Generative modeling has witnessed tremendous breakthroughs in recent years where diffusion models have emerged as a powerful tool for generating high-fidelity realistic images, videos, natural language content and even molecular data (Ho et al., 2020; Song et al., 2020; Bar-Tal et al., 2024; Wu et al., 2022). While diffusion models have demonstrated effectiveness in modeling complex and realistic data distributions, their successful application often hinges on following user-specific instructions in the form of images, text, bounding-boxes or downstream reward-functions. A common approach for *conditioning* diffusion models on user-specific input involves training them on data paired with fixed-

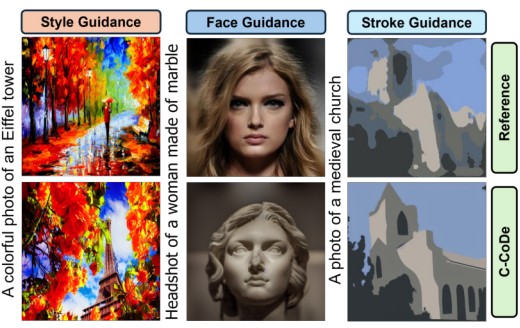

Figure 1: `C-CoDe` can flexibly generate high quality style, face and stroke guided images, while being considerably faster than most counterparts.

modality instruction signals in the form of descriptive text-prompts, segmentation maps, class-labels, etc. Another strategy for conditioning involves finetuning a pretrained diffusion model, either on a task-specific dataset or a reward-function. Finetuning is typically governed by reinforcement learning (RL), where the goal is to generate samples that optimize for a downstream reward-function while maintaining a low divergence with the pretraining data distribution. Despite their effectiveness, these conditioning strategies face their own set of challenges such as limited flexibility w.r.t. different instruction modalities, hindered generalizability to various domains due to their dependence on task-specific datasets, and high computational costs of training from scratch.

*Guidance*-based approaches keep the diffusion model intact and control its output by aligning its generative process to a reward function at inference-time; thus, offering potential remedies to the aforementioned challenges. Our proposed approach lies under this category. In this space, gradient-based guidance methods utilize gradients of the reward model at each diffusion denoising step to align the generated samples with the downstream task. Interesting follow-up works have addressed bias estimation challenges in computing gradients (Chung et al., 2023; Yu et al., 2023; Bansal et al., 2024b; He et al., 2024). Despite their flexibility to handle various downstream tasks, these approaches require memory-intensive gradient computation of differentiable guidance models. Staying under the premise of inference-time guidance, we propose a *gradient-free block-wise* guidance approach drawing inspiration from a related line of research in the context of language model (LM) alignment (Mudgal

et al., 2024), which capitalizes on the empirical strength of Best of $N$ (a.k.a. BoN) sampling (Gao et al., 2022; Mudgal et al., 2024), which is also theoretically shown to closely follow the optimal Kullback-Leibler (KL)-regularized objective (Yang et al., 2024). We introduce a simple block-wise controlled denoising (`CoDe`) method for diffusion models outperforming BoN at a fraction of its cost (much smaller $N$). Our end solution, termed as conditional controlled denoising (`C-CoDe`), incorporates adjustable noise conditioning on input images further optimizing `CoDe` from sampling efficiency perspective, as well as providing greater control over reward vs. divergence trade-off for more versatile generation. Our *key contributions* can be summarized as follows:

**I.** We propose an inference-time block-wise guidance approach (`CoDe`) which samples from an optimal KL-regularized objective. Building upon this base module, we further optimize it from sampling efficiency perspective and enhance it to offer adjustable reward-divergence trade-off (`C-CoDe`).

**II.** We assess the performance of the aligned diffusion model structurally for two case studies (Gaussian Mixture Model, and image generation), and three scenarios under image generation (style, face, and stroke guidance), by probing different aspects of the performance.

**III.** Our extensive (qualitative and quantitative) experimental results demonstrate that `C-CoDe` outperforms state-of-the-art baselines, while offering a balanced trade-off between reward alignment, prompt instruction following, and inference cost.

## 2 RELATED WORK

**Finetuning-based Alignment.** Prominent methods in this category typically involve either training a diffusion model to incorporate additional inputs such as category labels, segmentation maps, or reference images (Ho et al., 2021; Li et al., 2023; Zhang et al., 2023; Bansal et al., 2024a; Mou et al., 2024; Ruiz et al., 2023) or applying reinforcement learning (RL) to finetune a pretrained diffusion model to optimize for a downstream reward function (Prabhudesai et al., 2023; Fan et al., 2023; Wallace et al., 2023; Black et al., 2023; Gu et al., 2024; Lee et al., 2024; Uehara et al., 2024). While these approaches have been successfully employed to satisfy diverse constraints, they are computationally expensive. Furthermore, finetuning diffusion models is prone to "reward hacking" or "overoptimization" (Clark et al., 2024; Jena et al., 2024), where the model loses diversity and collapses to generate samples that achieve very high rewards. This is often due to a mismatch between the intended behavior and what the reward model actually captures. In practice, a perfect reward model is extremely difficult to design. As such, here we focus on inference-time guidance-based alignment approaches where these issues can be circumvented.

**Gradient-based Alignment.** There are two main divides within this category: (i) guidance based on a *value* function, and (ii) guidance based on a downstream *reward* function. In the first divide, a value function is trained offline using the noisy intermediate samples from the diffusion model. Then, during inference, gradients from the value function serve as signals to guide the generation process (Dhariwal & Nichol, 2021; Yuan et al., 2023). A key limitation of such an approach is that the value functions are specific to the reward model and the noise scales used in the pretraining stage. Thus, the value function has to be retrained for different reward models as well as base diffusion models. The second divide of methods successfully overcomes this by directly using the gradients of the reward function based on the approximation of fully denoised images using Tweedie's formula (Chung et al., 2022; 2023; Yu et al., 2023). Interesting follow-up research has explored methods to reduce estimation bias (Zhu et al., 2023; Bansal et al., 2024b; He et al., 2024) and to scale gradients for maintaining the latent structures learned by diffusion models (Guo et al., 2024). Despite such advancements, the need for differentiable guidance functions can limit the broader applicability of the gradient-based methods.

**Tree-Search-based Alignment.** Tree-search alignment has recently gained attention in the context of autoregressive language models (LMs), where it has been demonstrated that Best of $N$ (BoN) approximates sampling from a KL-regularized objective, similar to those used in reinforcement learning (RL)-based finetuning methods (Gui et al., 2024; Beirami et al., 2024; Gao et al., 2022). This approach facilitates the generation of high-reward samples while maintaining closeness to the base model. (Mudgal et al., 2024) demonstrates that the gap between Best of $N$ (BoN) and token-wise decoding (Yang & Klein, 2021) can be bridged using a block-wise decoding strategy. Inspired by this line of research, we propose a simple block-wise alignment technique (tree search with a fixed depth) that offers key advantages: (i) it preserves latent structures learned by diffusion models without

requiring explicit scaling adjustments, unlike gradient-based methods, and (ii) it avoids "reward hacking" typically associated with learning-based approaches. Concurrently, Li et al. (2024) propose a related method, called SVDD-PM, based on the well-known token-wise decoding strategy in the LM space. In contrast, we devise a block-wise strategy (CoDe, in Section 6) because it allows further control on the level of intervention, and offers a trade-off between divergence and alignment which is of primal interest in the context of guided generation. We further enhance our approach by introducing a noise-conditioned variant (C-CoDe in Section 4.2) to offer greater control over guidance signals and to further improve alignment.

## 3 PRELIMINARIES

### 3.1 DIFFUSION MODELS

An unconditional diffusion model estimates probability density $q(x)$ by learning to invert a forward diffusion process. The forward process is a Markov chain iteratively adding small amount of random noise to "clean" data point $x_0 \in \mathcal{X}$ sampled from $q(x)$ over $T$ steps. The noisy sample at step $t$ is given by $x_t = \sqrt{\bar{\alpha}_t} x_0 + \sqrt{1 - \bar{\alpha}_t} \epsilon_t$, where $\epsilon_t \sim \mathcal{N}(0, 1)$, $\alpha_t = 1 - \beta_t$, $\bar{\alpha}_t = \prod_{t=1}^{T} \alpha_t$, and $\beta_t \in (0, 1)_{t=1}^{T}$ is a variance schedule (Ho et al., 2020; Nichol & Dhariwal, 2021). The forward process can then be expressed as:

$$q(x_{1:T}|x_0) = \prod_{t=1}^{T} q(x_t|x_{t-1}), \quad q(x_t|x_{t-1}) = \mathcal{N}(x_t; \sqrt{1 - \beta_t} x_{t-1}, \beta_t \mathbf{I}). \tag{1}$$

Now, to estimate $q(x)$, the diffusion model $p_\theta$ learns the conditional probabilities $q(x_{t-1}|x_t)$ to reverse the diffusion process starting from a fully noisy sample $x_T \sim \mathcal{N}(0, 1)$ as:

$$p_\theta(x_0) = p(x_T) \prod_{t=1}^{T} p_\theta(x_{t-1}|x_t), \quad p_\theta(x_{t-1}|x_t) = \mathcal{N}(x_{t-1}; \mu_\theta(x_t, t), \beta_t \mathbf{I}), \tag{2}$$

where the variance is fixed at $\beta_t \mathbf{I}$, and only $\mu_\theta(x_t, t)$ is learned as

$$\mu_\theta(x_t, t) = \frac{1}{\sqrt{\alpha_t}} \left( x_t - \frac{1 - \alpha_t}{\sqrt{1 - \bar{\alpha}_t}} \epsilon_\theta(x_t, t) \right). \tag{3}$$

Here, $\epsilon_\theta$ is a neural network which attempts to predict the noise added to $x_{t-1}$ in the forward as:

$$\epsilon_\theta(x_t, t) \approx \epsilon_t = \frac{x_t - \sqrt{\bar{\alpha}_t} x_0}{\sqrt{1 - \bar{\alpha}_t}}. \tag{4}$$

Furthermore, using a conditioning signal $c$, diffusion models can be extended to sample from $p_\theta(x|c)$. The conditioning signal can take diverse forms, from text prompts and categorical information to semantic maps (Zhang et al., 2023). Our work uses a text-conditioned model, Stable Diffusion (Rombach et al., 2021), which has been trained on a large corpus consisting of $M$ image-text pairs $\mathcal{D} = \{(x^i, c^i)\}_{i=1}^{M}$ using a reweighed version of the variational lower bound (Ho et al., 2020) $\mathbb{E}_{t \sim [1,T], x_0, \epsilon_t} \left[ \|\epsilon_t - \epsilon_\theta(\sqrt{\bar{\alpha}_t} x_0 + \sqrt{1 - \bar{\alpha}_t} \epsilon_t, c, t)\|^2 \right]$ as optimization loss function.

### 3.2 KL-REGULARIZED OBJECTIVE

Consider we have access to a text-conditioned diffusion model $p_\theta(\cdot|c)$, which we refer to as the *base* model. Our goal is to obtain samples from the base model that optimize a downstream reward function $r(\cdot) : \mathcal{X} \to \mathbb{R}$, while ensuring that the sampled data points do not deviate significantly from $p_\theta$ to prevent degeneration in terms of image fidelity and diversity of the output samples (Ruiz et al., 2023). Thus, we aim to sample from a reward *aligned* diffusion model ($\pi_\theta$) that optimizes for a KL-regularized objective to satisfy both requirements. Let us start by defining some key concepts.

**Value function**. It captures the expected reward when decoding continues from a partially decoded sample $x_t$ given text prompt $c$ as:

$$V(x_t; p_\theta, c) = \mathbb{E}_{x_0 \sim p_\theta(x_0|x_t, c)}[r(x_0)]. \tag{5}$$

**Advantage function**. We can define a one-step advantage of using another text-conditioned diffusion model $\pi_\theta$ for optimizing the downstream reward as:

$$A(x_t; \pi_\theta, c) := \mathbb{E}_{x_{t-1} \sim \pi_\theta(x_{t-1}|x_t,\, c)} [V(x_{t-1}; p_\theta, c)] - \mathbb{E}_{x_{t-1} \sim p_\theta(x_{t-1}|x_t,\, c)} [V(x_{t-1}; p_\theta, c)]. \quad (6)$$

It is important to note that the advantage of the base model (when $\pi_\theta = p_\theta$) is 0. Thus, we aim to choose an *aligned* model $\pi_\theta$ to achieve a positive advantage over the base model.

**Divergence**. We further denote the KL divergence between the aligned model $\pi_\theta$ and the base model $p_\theta$ at each intermediate step $x_t$ as:

$$D(x_t; \pi_\theta, c) := KL\big[\pi_\theta(x_{t-1}|x_t,\, c) \,\|\, p_\theta(x_{t-1}|x_t,\, c)\big]. \quad (7)$$

**Objective**. Using Eq. 6 and 7, we can now formulate the KL-regularized objective as:

$$\pi_\theta^* = \arg\max_{\pi_\theta} \big[J_\lambda(x_t, \pi_\theta, c) := \lambda A(x_t; \pi_\theta, c) - D(x_t; \pi_\theta, c)\big], \quad (8)$$

where $\lambda \in \mathbb{R}^{\geq 0}$ trades off reward for drift from the base diffusion model $p_\theta$.

**Theorem 3.1.** *The optimal model $\pi_\theta^*$ for the objective formulated in Eq. 8 is*

$$\pi_\theta^*(x_{t-1}|x_t,\, c) \propto p_\theta(x_{t-1}|x_t,\, c)\, e^{\lambda V(x_{t-1}; p_\theta,\, c)}. \quad (9)$$

The proof of Theorem 3.1 is deferred to the Appendix A. A similar objective (or its variant) has been used in some learning-based methods (Prabhudesai et al., 2023; Fan et al., 2023; Wallace et al., 2023; Black et al., 2023; Gu et al., 2024; Lee et al., 2024) discussed in Section 2 for finetuning a diffusion model. However, contrary to the prior art, we use this objective for a guidance-based alignment. In Appendix B, we demonstrate that this can be achieved using Langevin dynamics (Welling & Teh, 2011), resulting in a generalized form of classifier guidance (Dhariwal & Nichol, 2021). A key limitation of such an approach is the need for a differentiable reward function. Therefore, we explore a sampling-based method for alignment with downstream rewards.

# 4 (CONDITIONAL) CONTROLLED DENOISING

Inspired by recent RL-based alignment strategies for LLM's (Yang & Klein, 2021; Mudgal et al., 2024), we propose a sampling-based guidance method to align a pretrained diffusion model ($p_\theta$) following the optimal solution described in Theorem 9 ($\pi_\theta^*$). First, we outline an approach to approximate the value function for intermediate noisy samples. Building on this approximation, we introduce our sampling-based alignment method coined as `CoDe`. We additionally introduce a variant of `CoDe`, termed as `C-CoDe`, by conditioning the initial noise on an input image provided by the user offering extra degrees of control and allowing for applications such as reference face/style conditioning (Bansal et al., 2024b) and stroke painting generation (Meng et al., 2021). Notably, this lowers the overall computational complexity substantially, by reducing the number of denoising steps as well as the number of samples, while achieving effective alignment in high-dimensional spaces.

**Approximation of the value function.** To compute the value function in Eq. 5 for an intermediate noisy sample $x_t$, it is necessary to compute the expectation over $x_0 \sim p_\theta(x_0|x_t)$. Note that for diffusion models such as DDPMs (Ho et al., 2020), the predicted clean sample $x_0$ can be estimated given an intermediate sample $x_t$ using Tweedie's formula (Efron, 2011) as follows:

$$\hat{x}_0 = \mathbb{E}[x_0|x_t] = \frac{x_t - \sqrt{1 - \bar{\alpha}_t}\, \epsilon_\theta(x_t, c, t)}{\sqrt{\bar{\alpha}_t}}. \quad (10)$$

By plugging Eq. 10 into Eq. 5, the value function can be approximated as:

$$V(x_t; p_\theta, c) = \mathbb{E}_{x_0 \sim p_\theta(x_0|x_t, c)}[r(x_0)] \geq r(\mathbb{E}[x_0|x_t]) = r(\hat{x}_0). \quad (11)$$

The benefit of such an approximation is that it circumvents the need for training a separate model to learn the value function, as is for instance adopted by DPS (Chung et al., 2023) and Universal Guidance (Bansal et al., 2024b).

**Best-of-N (BoN) sampling for diffusion models.** A naïve sampling-based approach for generating images from a diffusion model aimed at optimizing a downstream reward is Best-of-N (BoN). Here, first $N$ samples are obtained from the diffusion model by completely unrolling it out over $T$ denoising steps. Then, the most favorable image based on a value function is selected. Empirical evidence from the realm of large language models (LLMs) (Gao et al., 2022; Mudgal et al., 2024; Gui et al., 2024) suggests that BoN closely approximates sampling from the optimal solution presented in Theorem 3.1, which is theoretically corroborated by Yang et al. (2024).

## 4.1 BLOCK-WISE SAMPLING-BASED ALIGNMENT (CoDe)

Our objective is to achieve an improved alignment vs. divergence trade-off by sampling from the optimal solution presented in Theorem 3.1. Therefore, by taking advantage of the approximation in Eq. 11, we present a simple yet elegant sampling-based alignment method for diffusion models, termed as **Co**ntrolled **De**noising (CoDe) and outlined in Algorithm 1. CoDe integrates BoN sampling into the standard inference procedure of a pretrained diffusion model. However, instead of rolling out the entire diffusion model $N$ times and selecting the best sample, we opt for performing block-wise BoN. Specifically, for each block of $B$ steps, we unroll the diffusion

---

**Algorithm 1:** CoDe

**Require:** $p_\theta, T, N, B, x_T, c$
1  Initialize counter: $s = 1$
2  **for** $t \in [T-1, \cdots, 0]$ **do**
3     **if** $\mathrm{mod}(s, B) = 0$ **then**
4        Sample $N$ times over $B$ steps:
        $\{x_{t-1}^{(n)}\}_{n=1}^N \sim \prod_{i=t}^{t+B} p_\theta(x_{i-1}|x_i)$
5        Select the sample with maximum value:
        $x_{t-1} \leftarrow \underset{\{x_{t-1}^{(n)}\}_{n=1}^N}{\mathrm{argmax}}\ V(x_{t-1}^{(n)}; p_\theta, c)$
6     **end**
7     $s \leftarrow s + 1$
8  **end**
**Return:** $x_0$

---

model $N$ times independently (Algorithm 1, line 5). Then, based on the value function, select the best sample (Algorithm 1, line 6) to continue the reverse process till we obtain a clean image at $t = 0$. For the sake of brevity, we assume $T$ to be divisable by $B$; otherwise, we apply the same steps on a last but smaller block. Note that in terms of computational complexity, both BoN sampling and CoDe require the same number of inference steps. However, unlike BoN, CoDe introduces control at every block of $B$ steps, offering a more granular approach. A key advantage of CoDe is its ability to achieve similar alignment-divergence trade-offs while using a significantly lower value of $N$, as is demonstrated in Section 5.

## 4.2 C-CoDe: NOISE CONDITIONING FOR CoDe

When the reward distribution deviates significantly from the base distribution $p_\theta$, CoDe and any other sampling-based approach would require a relatively larger value of $N$ to achieve alignment. To tackle this, we introduce a variant of our method, termed as **C**onditional CoDe (C-CoDe), as described in Algorithm 2. In this variant, a reference target image $x_{\mathrm{ref}}$, such as a specific style or even stroke painting,

---

**Algorithm 2:** C-CoDe

**Require:** $p_\theta, T, N, B, \eta, x_{\mathrm{ref}}, c$
1  Sample conditional initial noise:
2     $\tau = \eta \times T$
3     $x_\tau = \sqrt{\bar{\alpha}_\tau} x_{\mathrm{ref}} + \sqrt{1 - \bar{\alpha}_\tau} z,\ z \sim \mathcal{N}(0, I)$
4  Sample using CoDe:
5     $x_0 \leftarrow \mathrm{CoDe}(p_\theta, \tau, N, B, x_\tau, c)$
**Return:** $x_0$

---

is provided as an additional conditioning input. Inspired by image editing techniques using diffusion (Meng et al., 2021; Koohpayegani et al., 2023), we add partial noise corresponding to only $\tau = \eta \times T$ steps of the forward diffusion process, instead of the full noise corresponding to $T$ steps (Algorithm 2, line 2 and 3). Then, starting from this noisy version of the reference image $x_\tau$, CoDe progressively denoises the sample for only $\eta \times T$ steps to generate the clean, reference-aligned image $x_0$ (Algorithm 2, line 4, 5). By conditioning the initial noise sample $x_\tau$ on the reference image $x_{ref}$, we can generate images $x_0$ that better incorporate the characteristics and semantics of the reference image $x_{\mathrm{ref}}$ while adhering to the text prompt $c$. As we demonstrate throughout our experimentation, threshold $\eta$ now provides an *extra knob* allowing the user to efficiently trade off divergence for reward. Here, the reward-conditioning of the generated image is inversely proportional to the value of $\eta$. Notably, adopting C-CoDe alleviates the need for a large number of samples $N$ for reward-aligned generation, where the reward distribution deviates considerably from the base distribution (e.g. in style guidance). It also results in compute efficiency, as is discussed in Section 6.

## 5 EXPERIMENTS

We analyze the performance of CoDe and C-CoDe, comparing them against a suite of existing state-of-the-art guidance methods. Unless otherwise mentioned, for all experiments, we use a pretrained Stable Diffusion version 1.5 (Rombach et al., 2021) as our base model, which is trained on the LAION-400M dataset (Schuhmann et al., 2021). As highlighted earlier, we strive to present meaningful comparative (both qualitative and quantitative) results across a variety of scenarios. For quantitative evaluations, we generate 50 images per setting (i.e., prompt-reference image pair) with 500 DDPM steps. To achieve this, we have used NVIDIA A100 GPUs with 80GB of RAM. Through extensive experiments, we aim to answer:

**[Q1].** Does (C-)CoDe achieve a better alignment-divergence trade-off compared to other baselines?

**[Q2].** How does (C-)CoDe perform across guidance tasks qualitatively and quantitatively?

**[Q3].** Does (C-)CoDe offer better image vs. text alignment compared to other baselines?

**Baselines.** We sub-select a set of widely adopted baselines from the literature. Recall that our goal is to sample from the optimal value of the KL-regularized objective, as outlined in Theorem 3.1. One approach to achieve this, as detailed in Appendix B, is using a gradient-based method with an approximated value function, as in DPS (Chung et al., 2023), which serves as our first baseline. Further, Universal Guidance (UG) (Bansal et al., 2024b), our second baseline, improves upon DPS by offering better gradient estimation. Another way to sample from Theorem 3.1 is by using a sampling-based approach such as in CoDe and C-CoDe. In this direction, we consider Best-of-N (BoN) (Gao et al., 2022) and SVDD-PM (Li et al., 2024) as our third and fourth baselines.

**Evaluation Settings and Metrics.** We consider two evaluation setting. **Setting I**: a prototypical 2D Gaussian Mixture Models (GMMs) in Section 5.1, as is also studied in (Ho et al., 2021; Wu et al., 2024); **Setting II**: widely adopted image based evaluations using Stable Diffusion in Section 5.2 across three scenarios: (i) style, (ii) face and (iii) stroke guidance. For Setting I, we present trade-off curves for expected reward versus KL-divergence for all baselines. For Setting II, since calculating KL-divergence in high-dimensional image spaces is intractable, we use Frechet Inception Distance (FID) (Heusel et al., 2017). To ensure we capture alignment w.r.t reference image (and avoid using the guidance reward itself) we borrow an image alignment metric commonly used in style transfer domain (Gatys et al. (2016); Yeh et al. (2020)), referred to as I-Gram here. Further, we assess prompt alignment using CLIPScore (Hessel et al., 2021), referred to as T-CLIP throughout the paper. Additionally, we consider Win-Rate (commonly adopted in the LM space) as yet another evaluation metric, where it reflects on the number of samples offering larger reward than the base model. To sum up, we consider expected reward, FID, I-Gram, T-CLIP, and Win-Rate.

## 5.1 Case Study I: Gaussian Mixture Models (GMMs)

To establish an in-depth understanding of the impact of the proposed methods, we start with a simple model/reward distribution as shown in Fig. 2 (top row). For the prior distribution, we consider a 2D Gaussian mixture model $p(\boldsymbol{x}_0) = \sum_{i=0}^{2} w_i \mathcal{N}(\boldsymbol{\mu}_i, \boldsymbol{\sigma}^2 \boldsymbol{I}_2)$, where $\sigma = 2$, $[\boldsymbol{\mu}_1, \boldsymbol{\mu}_2, \boldsymbol{\mu}_3] = [(5,3),(3,7),(7,7)]$, and $\boldsymbol{I}_d$ is an $d$-dimensional identity matrix. Additionally, we define the reward distribution as $p(r|\boldsymbol{x}) = \mathcal{N}(\boldsymbol{\mu}_r, \boldsymbol{\sigma}_r^2 \boldsymbol{I}_2)$ with $\boldsymbol{\mu}_r = [14,3]$ and $\boldsymbol{\sigma}_r = 2$. As can be seen in the figure, in this case and by design, reward distribution is far off the peak of the prior. For a different scenario, see Appendix C. Using the closed-form expressions for both prior and reward distributions in this setting, we compute the posterior distribution as $p(\boldsymbol{x}|r) = p(\boldsymbol{x})p(r|\boldsymbol{x})/Z$ where $Z$ is the normalizing constant. Note that this posterior corresponds to the optimal solution in Theorem 3.1 as $p(r|\boldsymbol{x}) \propto \exp(r(\boldsymbol{x}))$ with $r(\boldsymbol{x}) = -1/2(\boldsymbol{x} - \boldsymbol{\mu}_r)^T(\boldsymbol{x} - \boldsymbol{\mu}_r)$. Here, we train a diffusion model with a 3-layer MLP that takes as input $(\boldsymbol{x}_t, t)$ and predicts the noise $\boldsymbol{\epsilon}_t$.

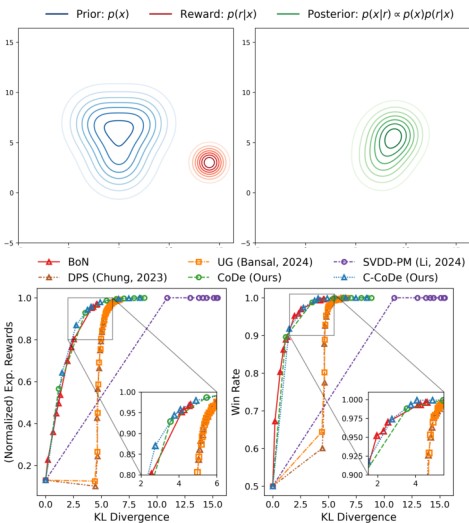

Figure 2: Setup (top row) and reward vs. divergence trade-off (bottom row) for Case Study I. C-CoDe offers highest reward at lowest divergence with much lower $N$ than BoN.

This model is trained over 200 epochs with $T = 1000$ denoising steps. Note that all other discussed baselines can straightforwardly be trained in this setting.

The results are illustrated in Fig. 2 (bottom row) where we plot the normalized expected reward and Win-Rate vs. KL-divergence for different value of $N \in [2, 500]$ as parameter. For the guidance-based methods DPS and UG, the guidance scale is varied between 1 and 50, whereas for the sampling-based methods BoN, SVDD, CoDe, and C-CoDe, the number of samples $N$ is varied between 2 and 500. As can be seen, for the expected reward, our proposed methods (CoDe and C-CoDe) offer the upper

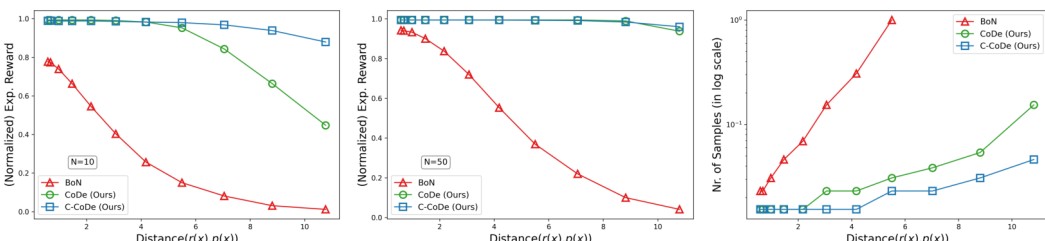

Figure 3: In contrast to BoN, proposed approaches are robust against increased distance between reward and prior distributions. `C-CoDe` (`CoDe`) achieves the same reward as BoN at much lower $N$.

bound of performance with a slight advantage over BoN. This order seems to be flipped when it comes to Win-Rate, which aligns with the observations from the realm of Language Models (LMs) (Beirami et al., 2024; Gui et al., 2024). In contrast, UG and DPS tend to exhibit higher KL divergence, as they often collapse to the mode of the reward distribution when the guidance scale is increased, leading to a reduction in diversity among the sampled data points, a phenomenon also noted in prior research (Sadat et al., 2024; Ho et al., 2021). In both scenarios, SVDD achieves a high expected reward (or Win Rate) but at the expense of significantly higher divergence, even for smaller values of $N$. In contrast, our methods offer flexibility, allowing users to balance the trade-off by adjusting parameters such as $N$ and $B$, as is demonstrated here and

Let us dive one step deeper into the performance of our proposed approaches and BoN. To this aim, in Fig. 3, we vary the distance between the mean of the reward and prior distributions, gradually shifting the reward further away. This is shown for $N = 10, 50$ in Fig. 3 where the expected reward sharply drops for BoN regardless of choice of $N$, whereas it drops less or remains almost intact for `CoDe` and `C-CoDe`, with N = 10 and 50, respectively. The key takeaway is that our proposed approach offers a consistently higher reward even when the prior and reward distributions are distant. To further probe this, we fix the reward and investigate with how many samples each method achieves the target reward. As can be seen on the right most figure, `C-CoDe` and `CoDe` meet this condition by outperforming significantly in terms of sample efficiency.

## 5.2 Case Study II: Image Generation with Stable Diffusion

We consider three commonly adopted guidance scenarios: style, stroke and face guidance. For each scenario, the rewards model is task specific as elaborated in the following. A text prompt as well as a reference image are used as guidance signals. A total of 33 generation settings (i.e., text prompt - reference image pairs) are used for evaluations in this section. Per setting, we generate 50 samples and estimate the evaluation metrics accordingly. On the qualitative side, to demonstrate the capacity of `C-CoDe` compared to other baselines, we illustrate a few generated examples across two reference images for two different text prompts. On the quantitative side, we evaluate the performance across all scenarios/settings combined for further statistical significance.

**Style guidance.** We guide image generation based on a reference style image (Bansal et al., 2024b; He et al., 2024; Yu et al., 2023). Following the reward model proposed in Bansal et al. (2024b), we use the CLIP image extractor to obtain embeddings for the reference style and the generated images. The cosine similarity between these embeddings is then used as the guidance signal. **Face guidance.** To guide the generation process to capture the face of a specific individual (as in (He et al., 2024; Bansal et al., 2024b)), we employ a combination of multi-task cascaded convolutional network (MTCNN) (Zhang et al., 2016) for face detection and FaceNet (Schroff et al., 2015) for facial recognition, which together produce embeddings for the facial attributes of the image. The reward is then computed as the negative $\ell_1$ loss between the face feature embeddings of the reference and generated images. **Stroke guidance.** A closely related scenario to style guidance is Stroke generation, where a high-level reference image containing only coarse colored strokes is used as reference (Cheng et al., 2023; Meng et al., 2021). The objective in this setting is to produce images that remain *faithful* to the reference strokes. To achieve this, similar to style guidance, we employ the CLIP image extractor to obtain embeddings from both the reference and generated images and compute the reward by measuring the cosine similarity between these embeddings.

**Qualitative Comparisons.** A comparative look across baselines, scenarios and settings is illustrated in in Figs. 4, 5 and 6. Let us start with style guidance in Fig. 4. As can be seen, `C-CoDe` shows

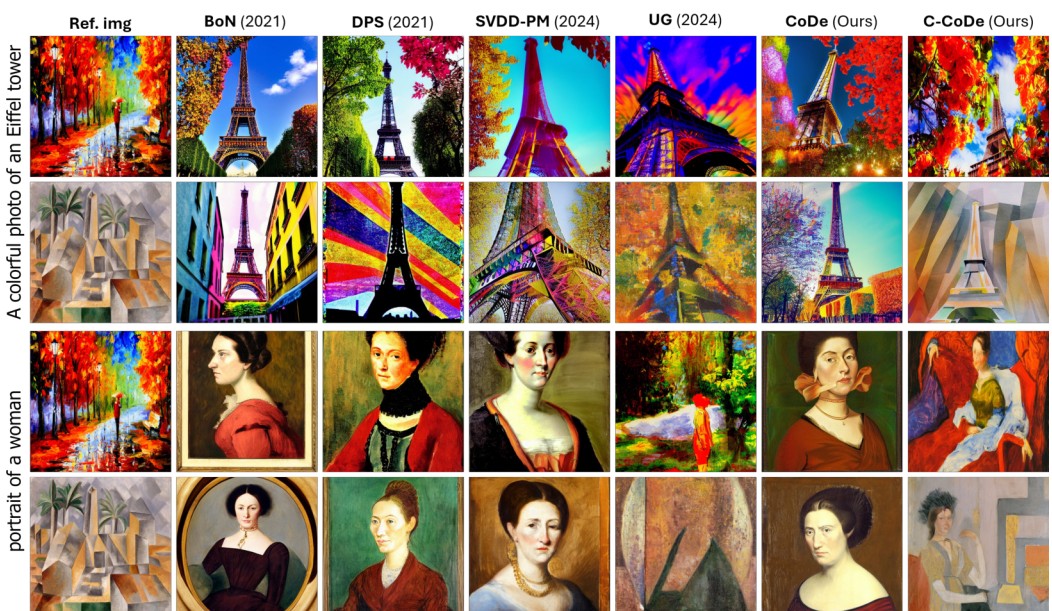

Figure 4: `C-CoDe` is a versatile approach presenting best alignment to the reference image, while adhering to the text prompt. The style alignment offered by `C-CoDe` outperforms other baselines by a margin in terms of quality and preserving nuances.

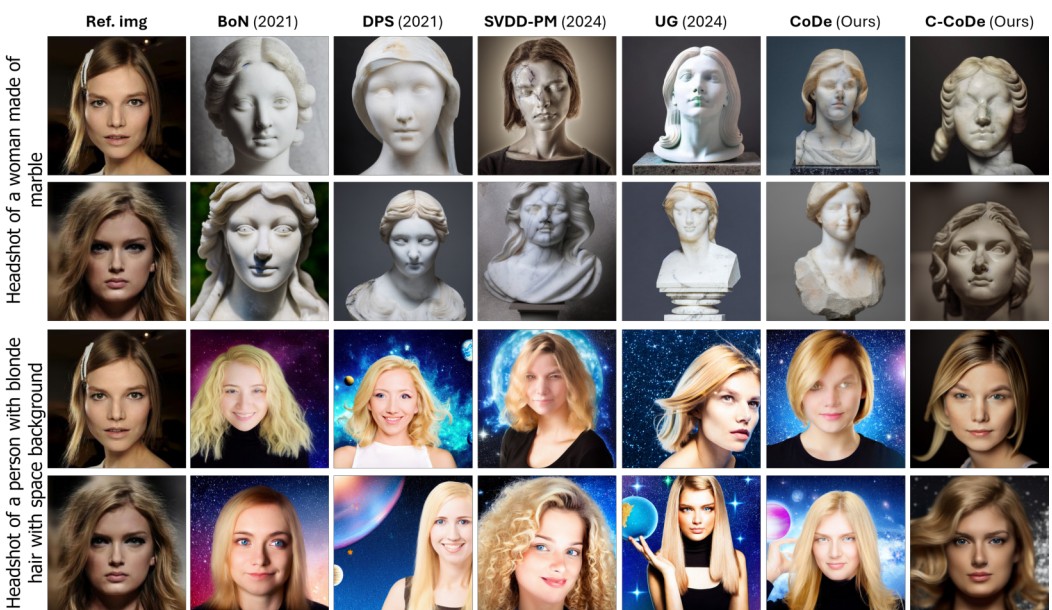

Figure 5: Same narrative as in Fig. 4 with `C-CoDe` outperforming other baselines by a margin.

versatility and superior performance in capturing the style of the reference image, regardless of the text prompt. Apart from UG, all other baselines (including our base module `CoDe` in certain cases), fail to capture the essence of the reference style. When it comes to alignment to the text prompt, however, UG seems to suffer to some extent with "woman" fading away in the bottom two rows. All other baselines tend to capture the text prompt predominantly and arguably fail to capture style. Note that from this angle `C-CoDe` outperforms UG by a noticeable margin, regardless of the reference image or the text prompt. Note that even our base module (`CoDe`) offers arguably similar results to those of SVDD-PM at the cost of much lower computational complexity (as is detailed in Tables 1). Further qualitative results for face and stroke guidance scenarios are summarized in Figs. 5 and 6.

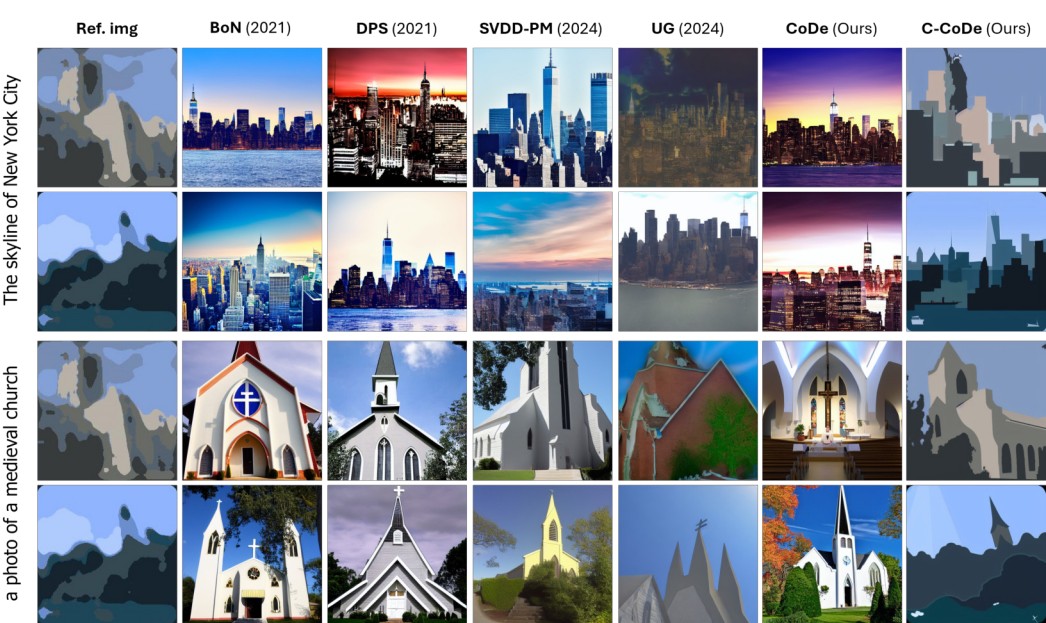

Figure 6: Same narrative as in Fig. 4 with `C-CoDe` outperforming other baselines by a margin.

Same narrative and observations extend here. The adherence of `C-CoDe` to the reference faces is worth highlighting. Same conclusions can drawn in the case of stroke guidance in Fig. 6 where no other baseline preserves the boundaries, color palette and nuances of the strokes as good as `C-CoDe`. The rest of the illustrations are self-explanatory.

**Quantitative Evaluations.** Table 1 summarizes the performance across all scenarios (including all settings) over four metrics: I-Gram, FID, T-CLIP and runtime (in second/image, and detailed Section 5.4). The reason why we use I-Gram (instead of expected reward per scenario) in our evaluations is because expected reward has been "seen" by the model

Table 1: Quantitative performance evaluation ($\pm$ std.).

| Method | FID ($\downarrow$) | I-Gram ($\uparrow$) | T-CLIP ($\uparrow$) | Runtime ($\downarrow$) |
|---|---|---|---|---|
| Base-SD (2021) | 1.0 | 1.0 | 1.0 | 1.0 |
| BoN (2022) | 1.19 | 1.07 ($\pm$ 0.004) | 0.99 ($\pm$ 0.001) | 18.90 ($\pm$ 0.01) |
| SVDD-PM (2024) | 1.42 | 1.24 ($\pm$ 0.02) | 0.98 ($\pm$ 0.004) | 99.10 ($\pm$ 0.08) |
| DPS (2023) | 1.14 | 1.12 ($\pm$ 0.01) | 0.98 ($\pm$ 0.004) | 5.82 ($\pm$ 0.02) |
| UG (2024b) | 2.91 | 1.86 ($\pm$ 0.03) | 0.85 ($\pm$ 0.005) | 87.92 ($\pm$ 0.03) |
| **CoDe (Ours)** | 1.17 | 1.30 ($\pm$ 0.009) | 0.99 ($\pm$ 0.001) | 34.63 ($\pm$ 0.04) |
| **C-CoDe (Ours)** | 3.00 | 3.19 ($\pm$ 0.05) | 0.87 ($\pm$ 0.006) | 23.82 ($\pm$ 0.03) |

throughout the guidance process. For more complete set of results, see Appendix D. We report scores across all metrics by normalizing them w.r.t. the base Stable Diffusion model (denoted by Base-SD). As can be seen, our base module `CoDe` offers performance gains in terms of image and text alignment (I-Gram and T-CLIP scores) while deviating lesser from the base model (FID score), compared to all baselines except UG. While at the same time `CoDe` is considerably faster than both SVDD-PM and UG. `C-CoDe` outperforms all other baselines in terms of image alignment while staying competitive in terms of text alignment. This is also corroborated qualitatively by Figs. 4, 5, 6, where `C-CoDe` incorporates the reference image semantics and the text prompt better than its counterparts across all image generation settings. When reference images differ considerably from the prior distribution (of Base-SD), better image alignment naturally comes at the cost of higher divergence (reward-divergence trade-off). While diverging as much as UG, `C-CoDe` achieves the highest overall image alignment with roughly $4\times$ faster runtime performance.

## 5.3 ABLATIONS

Fig. 7 investigates the impact of varying block size ($B$) and noise ratio ($\eta$) for `C-CoDe` on image vs. text alignment. For reference, `CoDe` and UG are also depicted. Here, different points per curve represent sweeping on their main parameter ($N = [5, 10, 20, 30, 40, 100]$ for (C-)CoDe, and guidance scale of $[1, 3, 6, 12, 24]$ for UG). On the left image, increasing block size seems to limit the image alignment performance; or put differently same performance at a much larger $N$. Regardless of block size, `C-CoDe` curves fall on top of UG indicating a superior overall performance. On the

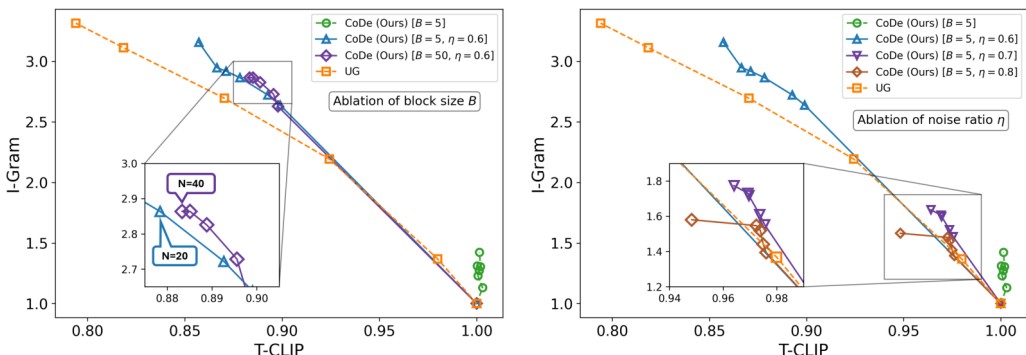

Figure 7: Ablation on the block size ($B$) and the noise ratio ($\eta$).

right, changing the noise ratio $\eta$ toward higher values, reduces the conditioning strength (as indicated also in (Meng et al., 2021; Koohpayegani et al., 2023)) resulting in lower image alignment capacity (I-Gram). Yet again, `C-CoDe` variants fall on top of the UG curve suggesting better image vs. text alignment performance. More detailed ablation studies are provided in Appendix D. Further note that the operation points with very low T-CLIP scores on UG curves ended up degenerating to the extent that images did not have anything in common with the text prompt, which was another consideration for choosing the best trade-off point.

## 5.4 COMPUTATIONAL COMPLEXITY.

We provide a comparative look at the complexity of the proposed approaches against the baselines. To this aim, we consider two aspects: (i) the number of inference steps, (ii) the number of queries to the reward model. We then measure the overall runtime complexity in terms of time (in sec.) required to generate one image. This is summarized in Table 2. From a runtime perspective, within the gradient-based guidance

Table 2: Computational complexity.

| Methods | Inf. Steps | Rew. Queries | Runtime [sec/img] |
|---|---|---|---|
| Base-SD (2021) | $T$ | - | 14.12 |
| BoN (2022) | $NT$ | $N$ | 266.77 |
| SVDD-PM (2024) | $NT$ | $NT$ | 1399.36 |
| DPS (2023) | $T$ | $T$ | 82.19 |
| UG (2024b) | $mKT$ | $mKT$ | 1241.47 |
| **CoDe (Ours)** | $NT$ | $NT/B$ | 489.00 |
| **C-CoDe (Ours)** | $NT$ | $rNT/B$ | 336.39 |

group, DPS is considerably faster across all three generation scenarios. This is due to the $m$ gradient and $K$ refinement steps used in UG, which are not used in DPS. Within the sampling based group, SVDD-PM, imposing token-wise agressive guidance, turns out to be an order of magnitude slower than BoN. `CoDe` asserting a block-wise guidance remains to be faster and more efficient than BoN as well as UG. `C-CoDe` further optimizes `CoDe` and offers a runtime of about $4\times$ faster than UG.

## 6 CONCLUDING REMARKS

We introduce a gradient-free block-wise inference-time guidance approach for diffusion models. By combining block-wise optimal sampling with an adjustable noise conditioning strategy, `C-CoDe` offers extra control over reward vs. divergence trade-off outperforming state-of-the-art baselines.

**Limitations and future work.** Diffusion models are computationally intensive; as such, extracting quantitative results on the performance of (inference-time) guidance-based alignment methods calls for massive resources, especially when ablating across numerous design parameters. We have used up to 32 NVIDIA A100's solely dedicated to the presented evaluation results. Yet, the 33 (most commonly adopted) settings we have experimented with to arrive at the numerical results of Table 1 is on the lower end of statistical significance. This calls for future work to carefully curate new benchmarks for evaluating these image generation tasks.

**Broader impact.** This work strives to take a meaningful step towards structurally analyzing the performance of diffusion models, in general, and provide simple alignment techniques. As such, we hope that it helps pave the way for a more in-depth study upon creation of a standard benchmark for this very purpose; something we have left as future work. However, we also caution against the blind use of the proposed techniques as the alignment methods are prone to reward over-optimization, which needs care especially in socially consequential applications.

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

## A    PROOFS

*Proof of Theorem 3.1.*

$$J_\lambda(x_t, \pi_\theta, c) = \mathbb{E}_{x_{t-1} \sim \pi_\theta} \left[ \lambda(V(x_{t-1}; p_\theta, c) - V(x_t; p_\theta, c)) + \log \frac{p_\theta(x_{t-1}|x_t, c)}{\pi_\theta(x_{t-1}|x_t, c)} \right] \quad (12)$$

$$= \mathbb{E}_{x_{t-1} \sim \pi_\theta} \left[ \log \frac{p_\theta(x_{t-1}|x_t, c) \; e^{\lambda(V(x_{t-1}; p_\theta, c) - V(x_t; p_\theta, c))}}{\pi(x_{t-1}|x_t, c)} \right] \quad (13)$$

$$= \mathbb{E}_{x_{t-1} \sim \pi_\theta} \left[ \log \frac{p_\theta(x_{t-1}|x_t, c) \; e^{\lambda V(x_{t-1}; p_\theta, c)}}{\pi_\theta(x_{t-1}|x_t, c)} + \log e^{\lambda V(x_t; p_\theta, c)} \right] \quad (14)$$

$$= \mathbb{E}_{x_{t-1} \sim \pi_\theta} \left[ \log \frac{p_\theta(x_{t-1}|x_t, c) \; e^{\lambda V(x_{t-1}; p_\theta, c)}}{\pi_\theta(x_{t-1}|x_t, c)} \right] + \lambda V(x_t; p_\theta, c) \quad (15)$$

Now, let

$$p_\lambda(x_{t-1}|x_t, c) := \frac{p_\theta(x_{t-1}|x_t, c) e^{\lambda V(x_{t-1}; p_\theta, c)}}{Z_\lambda(x_t, c)}, \quad (16)$$

where the normalizing constant $Z_\lambda(x_t, c)$ is given by

$$Z_\lambda(x_t, c) = \mathbb{E}_{x_{t-1} \sim p_\theta} \left[ p_\theta(x_{t-1}|x_t, c) e^{\lambda V(x_{t-1}; p_\theta, c)} \right]. \quad (17)$$

Putting it back in Eq. 15, we get

$$J_\lambda(x_t, \pi_\theta, c) = \mathbb{E}_{x_{t-1} \sim \pi_\theta} \left[ \log \frac{p_\lambda(x_{t-1}|x_t, c)}{\pi_\theta(x_{t-1}|x_t, c)} Z_\lambda(x_t, c) \right] + \lambda V(x_t; p_\theta, c) \quad (18)$$

$$= \mathbb{E}_{x_{t-1} \sim \pi_\theta} \left[ \log \frac{p_\lambda(x_{t-1}|x_t, c)}{\pi_\theta(x_{t-1}|x_t, c)} + \log Z_\lambda(x_t, c) \right] + \lambda V(x_t; p_\theta, c) \quad (19)$$

$$= \mathbb{E}_{x_{t-1} \sim \pi_\theta} \left[ \log \frac{p_\lambda(x_{t-1}|x_t, c)}{\pi_\theta(x_{t-1}|x_t, c)} \right] + \log Z_\lambda(x_t, c) + \lambda V(x_t; p_\theta, c) \quad (20)$$

$$= -\mathbb{E}_{x_{t-1} \sim \pi_\theta} \left[ \log \frac{\pi_\theta(x_{t-1}|x_t, c)}{p_\lambda(x_{t-1}|x_t, c)} \right] + \log Z_\lambda(x_t, c) + \lambda V(x_t; p_\theta, c) \quad (21)$$

$$= -KL(\pi_\theta(x_{t-1}|x_t, c) \parallel p_\lambda(x_{t-1}|x_t, c)) + \log Z_\lambda(x_t, c) + \lambda V(x_t; p_\theta, c) \quad (22)$$

Eq. 22 is uniquely maximized by $\pi_\theta^*(x_{t-1}|x_t, c) = p_\lambda(x_{t-1}|x_t, c)$.    □

## B    SAMPLING FROM OPTIMAL MODEL USING LANGEVIN DYNAMICS

Given the optimal policy given in Eq. 9, our goal is to now sample from $\pi^*$ instead of $p$. However, given only $p$, it is difficult to sample from this optimal policy. To overcome this problem, we look at the score-based sampling approach as in NCSN (Song & Ermon, 2019). Starting from an arbitrary point $x_T$, we iteratively move in the direction of $\nabla_{x_t} \log \pi^*(x_t)$, which is equivalent to $\nabla_{x_t} \log p_\lambda(x_t)$. We can derive an equivalent form:

$$p_\lambda(x_t) = \frac{p(x_t) e^{\lambda V(x_t)}}{Z_\lambda} \quad (23)$$

$$\log p_\lambda(x_t) = \log p(x_t) + \lambda V(x_t) - \log Z_\lambda \quad (24)$$

$$\nabla_{x_t} \log p_\lambda(x_t) = \nabla_{x_t} \log p(x_t) + \nabla_{x_t} \lambda V(x_t) - \nabla_{x_t} \log Z_\lambda \quad (25)$$

$$s_\lambda(x_t, t) = s_\theta(x_t, t) + \lambda \nabla_{x_t} V(x_t). \quad (26)$$

As the above derivation is limited to stochastic diffusion sampling, we leverage the connection between diffusion models and score matching (Song & Ermon, 2019):

$$\nabla_{x_t} \log p(x_t) = -\frac{1}{\sqrt{1 - \bar{\alpha}_t}} \epsilon_t. \quad (27)$$

**Similarity with classifier guidance**. Starting from an arbitrary point $x_T$, we iteratively move in the direction of $\nabla_{x_t} \log p(x_t|y)$. We can derive an equivalent form:

$$p(x_t|y) = \frac{p(y|x_t)p(x_t)}{Z} \tag{28}$$

$$\log p(x_t|y) = \log p(x_t) + \log p(y|x_t) - \log Z \tag{29}$$

$$\nabla_{x_t} \log p(x_t|y) = \nabla_{x_t} \log p(x_t) + \nabla_{x_t} \log p(y|x_t) - \nabla_{x_t} \log Z \tag{30}$$

$$s_\lambda(x_t|y,t) = s_\theta(x_t,t) + \nabla_{x_t} \log p(y|x_t). \tag{31}$$

## C  ADDITIONAL RESULTS FOR SETTING I

For the sake of completeness, we also study a variant of the GMM setting as discussed in Section 5.1, where the mean of the reward distribution is equal to the mean of one of the components in the prior distribution, as shown in Fig. 8. The prior distribution $p(\boldsymbol{x})$ is modelled as a 2-dimensional Gaussian mixture model (GMM) $p(\boldsymbol{x}_0) = \sum_{i=1}^{3} w_i \mathcal{N}(\boldsymbol{\mu}_i, \boldsymbol{\sigma}^2 \boldsymbol{I}_2)$, with $\sigma = 2$, $[\boldsymbol{\mu}_1, \boldsymbol{\mu}_2, \boldsymbol{\mu}_3] = [(5,3),(3,7),(7,7)]$, and $\boldsymbol{I}_d$ is an $d$-dimensional identity matrix, as shown in Fig. 2. All mixture components are equally weighted with, i.e., $w_1 = w_2 = w_3 = 0.33$. In contrast to the previous setup, we define the reward distribution as $p(r|\boldsymbol{x}) = \mathcal{N}(\boldsymbol{\mu}_r, \boldsymbol{\sigma}_r^2 \boldsymbol{I}_2)$ with $\boldsymbol{\mu}_r = [5,3]$ and $\boldsymbol{\sigma}_r = 2$. Based on this setup, we train a diffusion model $p_\theta(x)$ to estimate the prior distribution $p(\boldsymbol{x})$. For this we use a 3-layer MLP that takes as input $(\boldsymbol{x}_t, t)$ and predicts the noise $\boldsymbol{\epsilon}_t$. It is trained over 200 epochs with $T = 1000$ denoising steps. Then, we implement the baselines, `CoDe` and `C-CoDe`, to guide the trained diffusion model to generate samples with high likelihood under the reward distribution. Additionally, using the closed-form expressions for both the prior and reward distributions in this GMM configuration, we compute

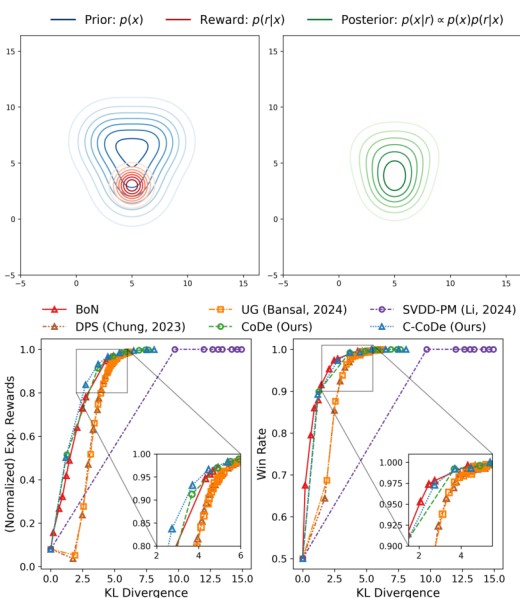

Figure 8: Setup (top row) and reward vs. divergence trade-off (bottom row) for another setting of Case Study I. `C-CoDe` offers highest reward at lowest divergence with much lower $N$ than BoN.

the posterior distribution as $p(\boldsymbol{x}|r) = p(\boldsymbol{x})p(r|\boldsymbol{x})/Z$ where $Z$ is the normalizing constant as shown in Fig. 8. This corresponds to the optimal solution in Theorem 3.1 as $p(r|\boldsymbol{x}) \propto \exp(r(\boldsymbol{x}))$ with $r(\boldsymbol{x}) = -1/2(\boldsymbol{x} - \boldsymbol{\mu}_r)^T(\boldsymbol{x} - \boldsymbol{\mu}_r)$.

In Fig. 8, we present the trade-off curves for normalized expected reward (or Win-Rate) versus KL divergence by adjusting the hyperparameters of the respective methods. For the guidance-based methods DPS and UG, the guidance scale is varied between 1 and 50, whereas for the sampling-based methods BoN, SVDD, `CoDe`, and `C-CoDe`, the number of samples $N$ is varied between 2 and 500. Similar to the results in Section 5.1, we observe `C-CoDe` and `CoDe` achieve the most favorable trade-off between normalized expected reward and KL divergence, with BoN performing closely behind. In the case of Win-Rate vs. KL divergence, BoN demonstrates the best trade-off, consistent with findings from the literature on Language Model (LM) alignment. Furthermore, guidance-based methods tend to exhibit higher KL divergence, as they often collapse to the mode of the reward distribution when the guidance scale is increased, leading to a reduction in diversity among the sampled data points. In both scenarios, SVDD achieves a high expected reward or win rate but at the expense of significantly increased divergence, even for smaller values of $N$. Whereas `CoDe` and `C-CoDe` offer the widely sought-after flexibility, allowing users to balance the trade-off by adjusting parameters such as $N$ and $B$.

# D ADDITIONAL RESULTS FOR SETTING II

Here, we provide further details about the quantitative evaluations summarized in Table 1 and computational complexity analysis in Table 2.

**Further details on evaluation metrics.** For computing I-Gram, we utilize VGG (Simonyan & Zisserman, 2014) Gram matrices of the reference and generated images to measure image alignment across all scenarios/settings, as commonly followed in the literature (Somepalli et al., 2024; Gatys et al., 2016; Yeh et al., 2020). Specifically, these are computed using the last layer feature maps of an ImageNet-1k pretrained VGG backbone (Simonyan & Zisserman, 2014). For face guidance, we utilize the last layer feature maps of an InceptionResNetV1 pretrained on the VGGFace2 dataset (Parkhi et al., 2015) in order to build the gram matrix. Image alignment between a reference, generated image pair is then measured by computing the dot product of their gram matrices. Further, we report a recently proposed CLIP-based Maximum Mean Discrepancy (CMMD) (Jayasumana et al., 2024) as a divergence measure. It overcomes the drawback of FID stemming from the underlying Gaussian assumption in the representation space of the Inception model (Szegedy et al., 2015).

**Quantitative performance.** In this section, we break down the quantitative performance of all methods across the three different scenarios of style, face and stroke guidance. We summarize the results in Tab. 3, 4, 5 with the first row corresponding to the base Stable Diffusion model and Rew. indicating the reward metric used for guiding the diffusion model

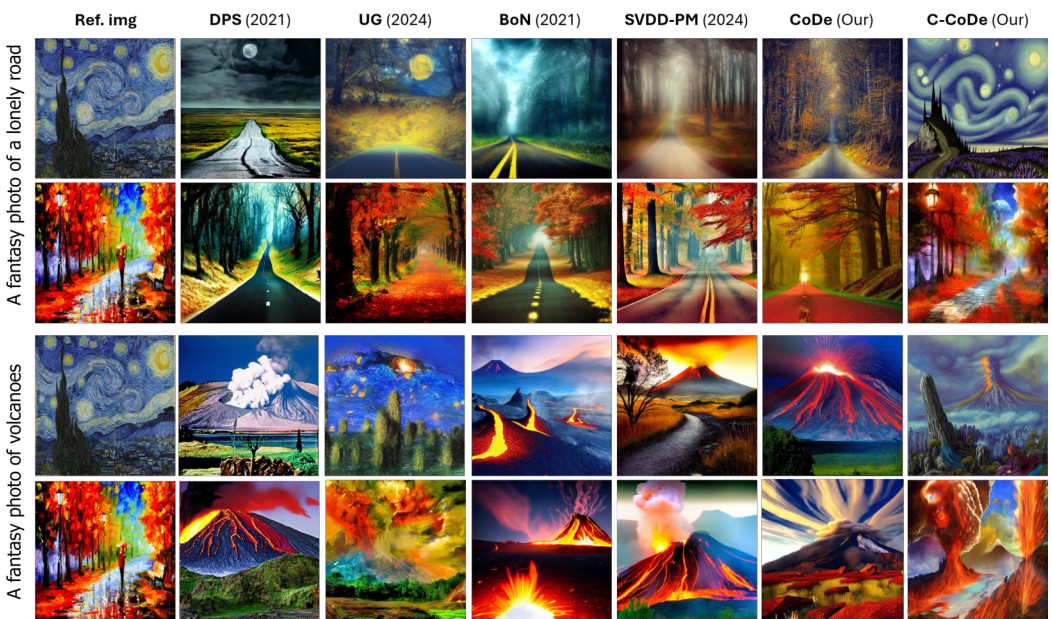

Figure 9: Quality evaluation across methods for style guidance

**Style Guidance**. The results are summarized in Table 3. Compared to the sampling-based guidance counterpart BoN, CoDe achieves a higher reward at the cost of slightly higher divergence (FID and CMMD). Yet, with a slightly smaller reward CoDe offers a better performance than UG and SVDD-PM across FID, CMMD and T-CLIP. The overall highest reward is

Table 3: Quantitative metrics for style guidance.

| Method | R1: Style Guidance | | | | |
|---|---|---|---|---|---|
| | Rew. (↑) | FID (↓) | CMMD (↓) | T-CLIP (↑) | I-Gram (↑) |
| Base-SD (2021) | 1.0 | 1.0 | 1.0 | 1.0 | 1.0 |
| BoN (2022) | 1.14 | 1.30 | 2.25 | 0.99 | 1.1 |
| SVDD-PM (2024) | 1.44 | 1.81 | 10.93 | 0.99 | 1.6 |
| DPS (2023) | 1.22 | 1.29 | 5.46 | 0.99 | 1.2 |
| UG (2024b) | 1.39 | 4.27 | 91.13 | 0.82 | 2.9 |
| **CoDe(Ours)** | 1.34 | 1.49 | 7.40 | 1.0 | 1.6 |
| **C−CoDe(Ours)** | 1.52 | 3.64 | 84.45 | 0.86 | 3.4 |

obtained by C-CoDe, which naturally comes with higher FID and CMMD scores. However, note that the divergence of C-CoDe is smaller than UG, the second-best method in this setting. This is also illustrated in Fig. 10 where C-CoDe consistently outperforms UG in terms of image alignment

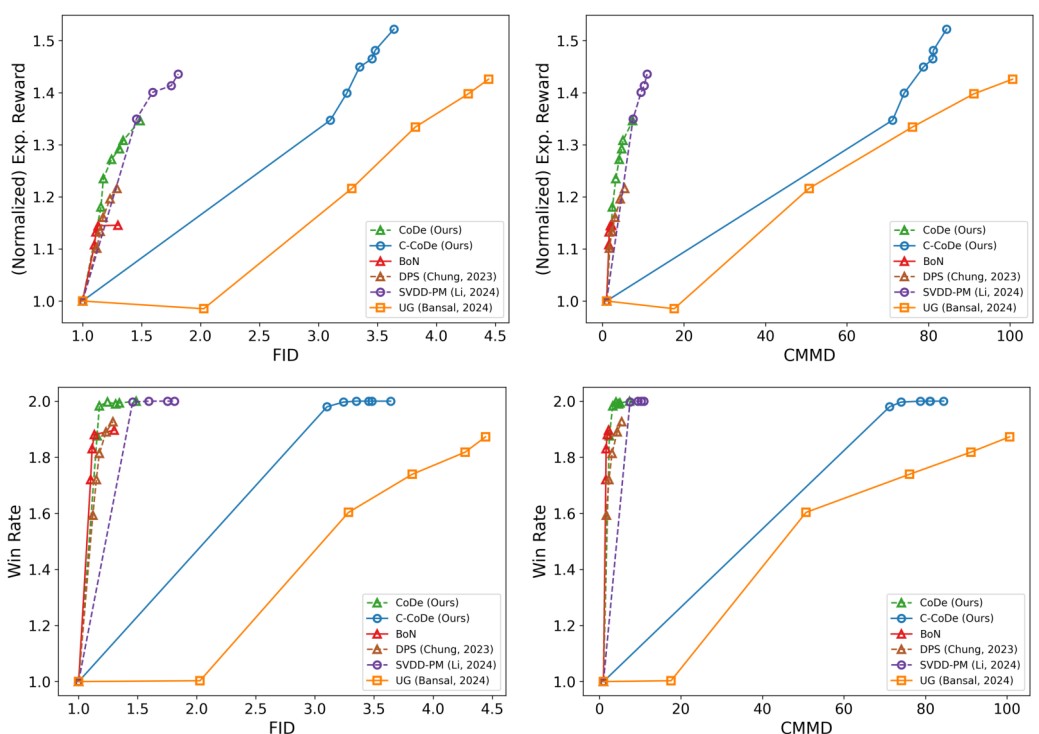

Figure 10: Reward vs. divergence trade-off curves for style guidance.

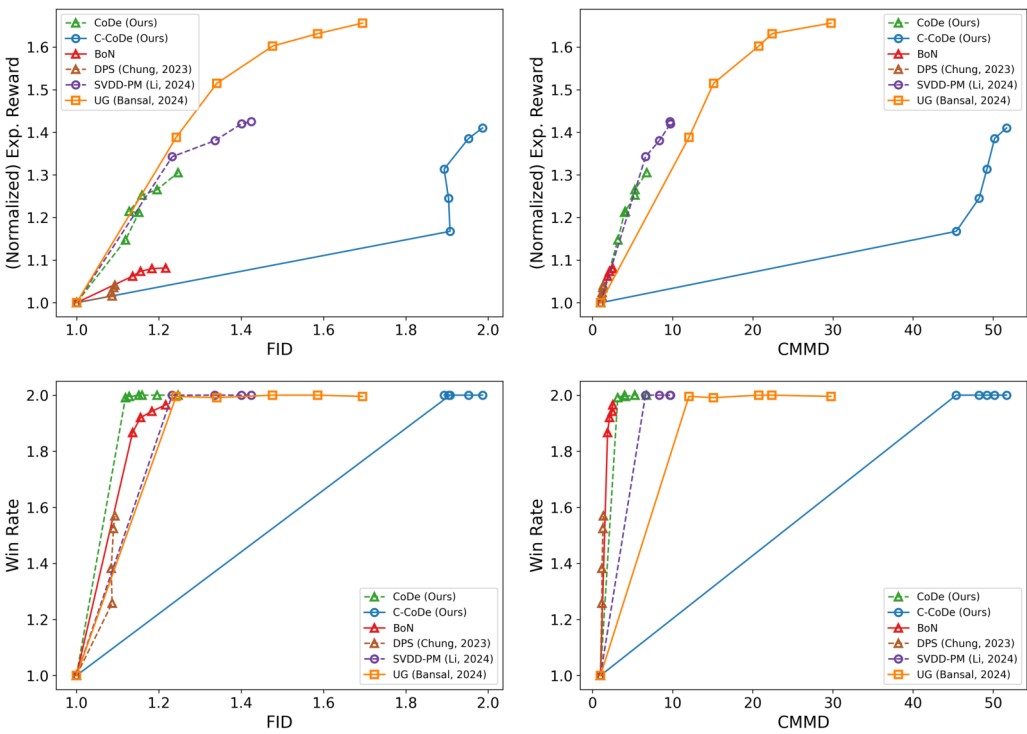

Figure 11: Reward vs. divergence trade-off curves for face guidance.

(normalized expected reward as well as win rate), while also offering lesser divergence w.r.t. both FID and CMMD as compared to UG.

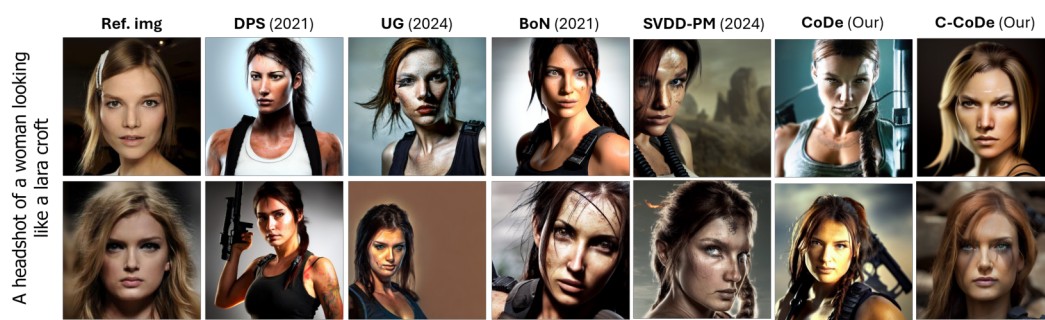

Figure 12: Quality evaluation across methods for style guidance

**Face Guidance**. We summarize the results in Table 4. As the rewards are negative, we first compute the negative log of the reward values and then normalize it with respect to the base. Compared to BoN, CoDe provides higher rewards with slightly higher divergence (FID and CMMD). Although SVDD-PM achieves slightly higher rewards, CoDe provides better

Table 4: Quantitative metrics for face guidance.

| Method | R2: Face Guidance | | | | |
|---|---|---|---|---|---|
| | Rew. (↑) | FID (↓) | CMMD (↓) | T-CLIP (↑) | I-Gram (↑) |
| Base-SD (2021) | 1.0 | 1.0 | 1.0 | 1.0 | 1.0 |
| BoN (2022) | 1.08 | 1.22 | 2.52 | 0.99 | 1.0 |
| SVDD-PM (2024) | 1.42 | 1.42 | 9.67 | 0.97 | 0.74 |
| DPS (2023) | 1.04 | 1.09 | 1.36 | 0.99 | 1.03 |
| UG (2024b) | 1.66 | 1.69 | 29.76 | 0.86 | 1.06 |
| **CoDe(Ours)** | 1.30 | 1.25 | 6.76 | 0.98 | 0.91 |
| **C−CoDe(Ours)** | 1.5 | 1.86 | 42.40 | 0.88 | 1.91 |

performance than UG, SVDD-PM and in terms of FID, CMMD and T-CLIP. Additionally, C-CoDe provides competitive results as compared to UG, which is the second-best method while offering better prompt alignment as reflected in a higher T-CLIP score. We draw similar conclusions from the reward vs. divergence curves presented in Fig. 11, where C-CoDe achieves competitive rewards but on-par win-rates as compared to UG, at the cost of slightly higher FID and CMMD scores.

**Stroke**. As shown in Table. 5, among the sampling-based methods, CoDe provides better results than BoN in terms of expected reward and FID while maintaining the same T-CLIP score. Although UG and SVDD-PM offer higher rewards, CoDe offers lower divergence (FID and CMMD) and better T-CLIP scores. Overall, we observe that C-CoDe has the highest rewards while offering competitive FID, CMMD and T-CLIP.

Table 5: Quantitative metrics for stroke generation.

| Method | R3: Stroke Generation | | | | |
|---|---|---|---|---|---|
| | Rew. (↑) | FID (↓) | CMMD (↓) | T-CLIP (↑) | I-Gram (↑) |
| Base-SD (2021) | 1.0 | 1.0 | 1.0 | 1.0 | 1.0 |
| BoN (2022) | 1.25 | 1.05 | 4.5 | 0.99 | 1.12 |
| SVDD-PM (2024) | 1.56 | 1.04 | 12.0 | 0.99 | 1.38 |
| DPS (2023) | 1.34 | 1.04 | 14.0 | 0.97 | 1.13 |
| UG (2024b) | 1.55 | 2.78 | 78.0 | 0.88 | 1.63 |
| **CoDe(Ours)** | 1.41 | 0.78 | 6.5 | 0.99 | 1.38 |
| **C−CoDe(Ours)** | 1.75 | 3.50 | 178.5 | 0.87 | 4.25 |

**Computation Complexity.** We present a breakdown of the computational complexities of all baselines across each of the guidance scenarios. DPS is considerably faster across all three generation scenarios among the gradient-based guidance methods. This is due to the $m$ gradient and $K$ refinement steps used in UG, which are not used in DPS. The difference is more pronounced in the case of style-

Table 6: Computational Complexity

| Methods | Inf. Steps | Rew. Queries | Runtime [sec/img] | | |
|---|---|---|---|---|---|
| | | | Style | Face | Stroke |
| Base-SD 2021 | $T$ | - | 14.12 | 14.12 | 14.12 |
| BoN 2022 | $NT$ | $N$ | 266.02 | 268.43 | 265.86 |
| SVDD-PM 2024 | $NT$ | $NT$ | 1168.74 | 1859.67 | 1169.68 |
| DPS 2023 | $T$ | $T$ | 62.52 | 122.21 | 61.83 |
| UG 2024b | $mKT$ | $mKT$ | 1588.41 | 543.12 | 1592.89 |
| CoDe (Ours) | $NT$ | $NT/B$ | 441.81 | 583.12 | 442.08 |
| C-CoDe (Ours) | $NT$ | $rNT/B$ | 331.42 | 403.19 | 274.56 |

and stroke guidance as UG uses a higher number of gradient steps $m$. Further, among the sampling-based approaches, SVDD-PM is slower than BoN in order of magnitude as it applies token-wise guidance. On the contrary, our block-wise approach C-CoDe is more efficient than UG and SVDD-PM and closely follows BoN.

## E  MISCELLANEOUS RESULTS

In this section, we illustrate several additional generated images across all baselines and guidance scenarios. We also provide additional results for `C-CoDe` across various different reference images and text prompt pairs, that are different from the ones already explored in the main manuscript, as illustrated in Fig

To broaden the understanding of our proposed approach `C-CoDe`, we utilize only the noise-conditioning aspect of `C-CoDe` to generate multiple images across all the style guidance (reference image, text prompt) settings. As can be seen in Figs. 13 only using reference image noise conditioning can also be used as a naive baseline for guided image generation. However, it is to be noted that using `CoDe` in conjunction with noise-conditioning, as demonstrated with `C-CoDe`, renders more sophisticated results in terms of capturing the nuances and subtleties of the reference image, while incorporating the semantics of the text prompt.

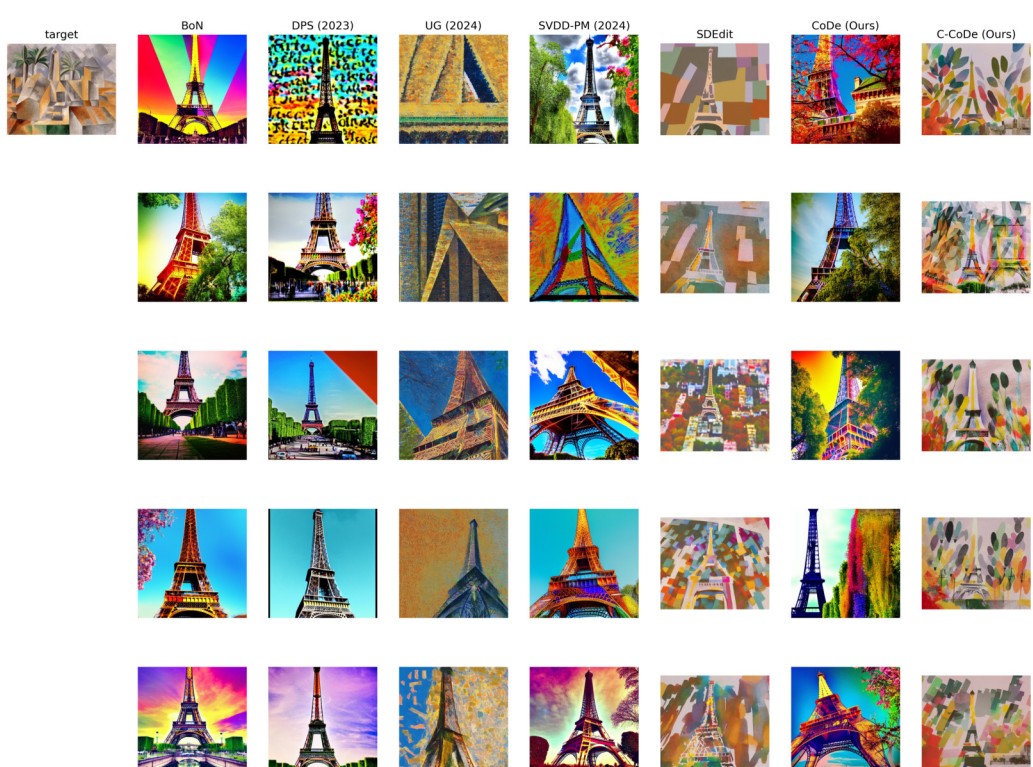

Figure 13: Multiple generated samples for the text prompt `A colorful photo of eiffel tower`.

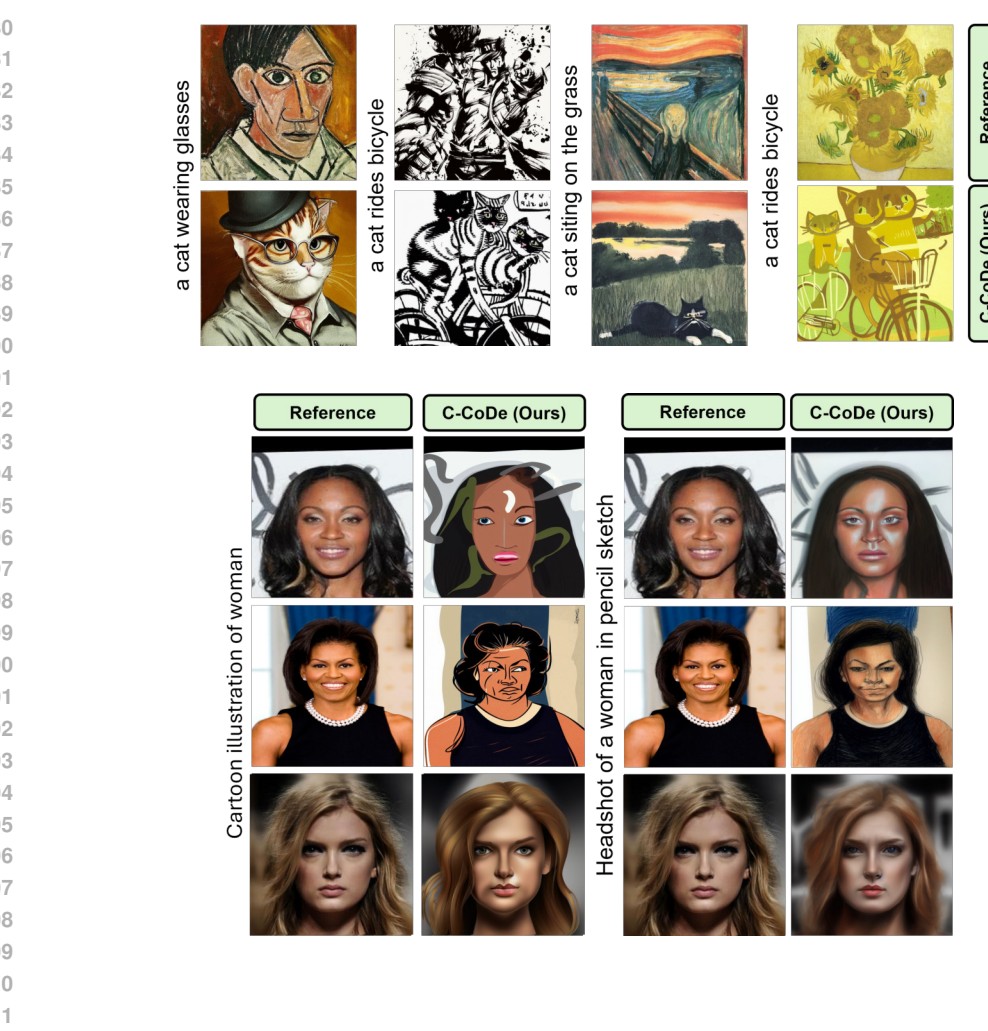

