# OpenReview forum: "Controlled Denoising For Diffusion Models"
_ICLR.cc/2025/Conference — ICLR 2025 Conference Withdrawn Submission_

### Official Review · Reviewer_Cjdo · 2024-10-23

**Soundness:** 2
**Presentation:** 3
**Contribution:** 1
**Rating:** 5
**Confidence:** 3

**Summary:**

Summary:

The authors propose a guidance approach for diffusion models which does not require a differentiable guidance function. Results are presented with respect to a synthetic and several visual tasks, including style guidance, face guidance and stroke guidance.

**Strengths:**

Strengths:

The paper is well-structured and generally easy to read. The authors perform ablation studies to test the influence of several of their design choices (e.g. block size), which is useful in elucidating the method and its efficacy under different settings of the same.

**Weaknesses:**

Weaknesses:

The work struggles to place itself in the current literature, and should include a much wider breadth of recent methods that aim to solve the very same problem. The claims seem also quite inflated (e.g. better performance by a margin with respect to concurrent methods), and not appropriately justified. A more thorough evaluation of current methods with respect to the proposed solution, as well as benchmarks with the same would significantly strengthen the paper.

**Questions:**

Questions:

- The related work section misses several relevant works, including unbiased sampling methods within the finetune literature (e.g. RTB \[1\]) and SMC-based methods  (e.g. \[2\] and \[3\]). These are also closely related to the objective explored in this paper (theorem 3.1). The authors should discuss the main differences and trade offs.
- It is not clear to me why eq (11) should generally be true. Can you elucidate?
- How does the unreliability of estimating r(x_0) for x_0 estimated at high temperatures affect sampling?
- There are several questions with the C-CoDe variant. For example, at times we might have a posterior which deviates significantly from the prior (given the condition), and no reference image to use, only a reward. There does not seem to be a principled way to take advantage of the tails of the distribution in the prior model.
- The Case study in 5.1 is much too simplistic to showcase what has been proposed. The gaussians are close to each other and the energy is simple to approximate. It is also hard to observe phenomena like mode collapse. It would be worthwhile to have a simple, yet multimodal, distribution (e.g. a posterior based on a Gaussian mixture model prior with very separate Gaussians), to at least test some coverage.
- Figure 2: The results are generally not very compelling. It would be good to have some idea of the statistical significance of these improvements. Upper figures are missing the x-tick labels.
- Figure 3, ticks/labels are too small to read
- Figure 4 and 5 and 6: the truthfulness of the sentiment in the caption is very subjective. The results in Table 1 do not overall support this claim. For example, C-CoDe achieves highest I-gram but is far from performing well in terms of FID. Tables 3, 4, and 5 in the appendices show similar trends. It would be worthwhile to know how the baselines were run. How was FID computed? How was the std computed, and why are the FID values missing it?
- DPS is sensitive to the scale, and would typically achieve very high rewards (likely at the expense of FID or other similar metrics); how was the scale tuned for the experiments?

\[1\] Venkatraman, S., Jain, M., Scimeca, L., Kim, M., Sendera, M., Hasan, M., Rowe, L., Mittal, S., Lemos, P., Bengio, E. and Adam, A., 2024. Amortizing intractable inference in diffusion models for vision, language, and control. _arXiv preprint arXiv:2405.20971_.

\[2\] Wu, L., Trippe, B., Naesseth, C., Blei, D. and Cunningham, J.P., 2024. Practical and asymptotically exact conditional sampling in diffusion models. _Advances in Neural Information Processing Systems_, _36_.

\[3\]  Cardoso, G., Le Corff, S. and Moulines, E., 2023. Monte Carlo guided Denoising Diffusion models for Bayesian linear inverse problems. In The Twelfth International Conference on Learning Representations.

---

### Official Review · Reviewer_Kg9h · 2024-10-30

**Soundness:** 1
**Presentation:** 2
**Contribution:** 1
**Rating:** 3
**Confidence:** 4

**Summary:**

The paper proposes a method for guiding the generation process in diffusion models. Inspired by techniques from large language models (LLMs), the paper proposes to use Best-of-N (BoN) sampling, which should circumvent the need for fine-tuning or differentiable guidance functions (e.g., classifier guidance). Specifically, the approach samples $N$ images and selects the one that best meets the objective of the guidance function. For more precise control with smaller values of $N$, the method introduces CoDe, which denoises each sample incrementally in blocks of $B$ steps rather than fully to $x_0$. To improve fidelity to the reference image, the method essentially incorporates SDEdit as a preprocessing step, resulting in an approach named C-CoDe. The paper presents experiments on a 2D toy setting and stable diffusion.

**Strengths:**

1. The paper tackles an important issue of guiding the generation process of diffusion models.
2. The paper uses an already established idea from NLP and attempts to apply it to diffusion models, which is an interesting approach.
3. The idea of using BoN has theoretical motivation.
4. The paper is well written and easy to follow.

**Weaknesses:**

In my view, this paper faces two main issues: (1) limited novelty, and (2) insufficient empirical results that fail to substantiate the proposed method’s contributions.

The proposed approach essentially combines two straightforward ideas: (1) Best-of-N (BoN) sampling, and (2) SDEdit. While simplicity can be a strength, this algorithm is rather trivial, consisting mainly of repeated sampling (BoN) and an effective initialization (SDEdit). Simple methods that aim to provide a strong baseline should convincingly outperform more complex alternatives, but this paper falls short, delivering weak results with key baseline comparisons omitted.

The experimental results, as shown in Figures 4, 5, 6, and Table 1, suggest that C-CoDe is a critical component. However, C-CoDe merely involves running SDEdit prior to CoDe. Consequently, SDEdit should serve as a primary baseline. A single qualitative comparison to SDEdit appears on the last page of the appendix (Appendix E, Figure 13), where SDEdit arguably performs slightly better. I believe SDEdit should be presented as a standalone baseline in every table and figure. Additionally, a C-BoN baseline (SDEdit + BoN) would provide an essential comparison, similar to C-CoDe.

Regarding results, while C-CoDe occasionally produces relevant outputs (e.g., Figure 4 rows 1–2 and Figure 6), these are likely due to the influence of SDEdit. The quantitative results are also unconvincing, with C-CoDe ranking near the bottom for FID and T-CLIP. Although it scores well on I-GRAM, this is difficult to interpret without an SDEdit baseline, and the remaining I-GRAM scores fall within expected noise levels. Furthermore, SDEdit is likely faster than C-CoDe.

For these reasons, I recommend rejecting the paper from ICLR.

**Questions:**

1. When comparing methods, are all the seeds fixed across the baselines (i.e., do all of them get the exact same initial noise)?
2. Why use I-Gram and not similarity between the CLIP Image embeddings?
3. What guidance scale did you use for UG in the figures? The ablation in Fig. 7 suggests it should perform similar to C-CoDe in terms of I-Gram but this is not the case depicted in Figs. 4,5,6.

---

### Official Review · Reviewer_xiW2 · 2024-11-01

**Soundness:** 3
**Presentation:** 3
**Contribution:** 2
**Rating:** 5
**Confidence:** 3

**Summary:**

The paper introduces Conditional Controlled Denoising (C-CoDe), a gradient-free method for aligning diffusion models with downstream tasks without model finetuning or differentiable guidance functions. Through block-wise sampling and conditioning on reference images, C-CoDe achieves efficient reward alignment, outperforming existing baselines.

**Strengths:**

1. The paper is well-written and easy to follow.
2. The proposed method is applicable to a wide range of downstream tasks without the added cost of retraining or requiring differentiable guidance functions.

**Weaknesses:**

1. The experiments lack comparison with relevant baseline methods, though several related studies are refered in Section 2, such as, MPGD [1], FreeDoM [2], SDEdit[3].
2. The proposed method incurs significantly higher runtime costs compared to Base-SD (14.12 sec/img), with C-CoDe (336.39 sec/img) being 23.82 times slower, which limits its practicality in real-world applications. Incorporating faster sampling methods, such as DDIM [4] or DPM-Solver [5], would help assess whether the proposed approach remains compatible with efficient sampling techniques.
3. The three applications presented—style guidance, face guidance, and stroke guidance—do not fully highlight the limitations of alternative methods. The authors could explore more representative scenarios, such as cases where differentiable guidance functions are not feasible, to better illustrate the advantages of the proposed method.
4. Hyperparameter settings for the proposed method and baseline comparisons are missing.

[1] He, Yutong, et al. "Manifold preserving guided diffusion." arXiv preprint arXiv:2311.16424 (2023).

[2] Yu, Jiwen, et al. "Freedom: Training-free energy-guided conditional diffusion model." Proceedings of the IEEE/CVF International Conference on Computer Vision. 2023.

[3] Meng, Chenlin, et al. "Sdedit: Guided image synthesis and editing with stochastic differential equations." arXiv preprint arXiv:2108.01073 (2021).

[4] Song, Jiaming, Chenlin Meng, and Stefano Ermon. "Denoising diffusion implicit models." arXiv preprint arXiv:2010.02502 (2020).

[5] Lu, Cheng, et al. "Dpm-solver: A fast ode solver for diffusion probabilistic model sampling in around 10 steps." Advances in Neural Information Processing Systems 35 (2022): 5775-5787.

**Questions:**

Please see weaknesses.

---

### Official Review · Reviewer_gjWe · 2024-11-03

**Soundness:** 3
**Presentation:** 3
**Contribution:** 2
**Rating:** 5
**Confidence:** 3

**Summary:**

The work proposes a guidance method based on Best-of-N. Given an objective function for downstream task, the method divide the sampling timesteps into B blocks, at each block, $N$ noisy samples are sampled from diffusion models and the best one based on the objective function is selected. In order to limit $N$, Conditional CoDE is proposed so that conditional reference information is input into the sampling process.

**Strengths:**

1. The paper is well written
2. The structure is clear

**Weaknesses:**

1. The motivation seems to be not clear. Since the same objective can be found in learning-based approaches, the paper does not state the main advantages of using the guidance compared to fine-tuning the downstream tasks. The comparison between guidance and learning-based approaches is necessary since, after fine-tuning, the fine-tuned models' sampling processes require much less computation than the proposed method.
2. The proposed method is actually a special version of BoN; the main difference is it cuts the sampling process into B blocks. Although it can be understood that this scheme helps to reduce the accumulated errors, it can also be interpreted as a "greedy" scheme, which will result in a "local minimum".
3. The connection between CoDe and the improved trade-off between alignment and divergence is not clearly explained.
4.  The sampling time of CoDe will be much more significant than BoN.
5. The motivation for introducing C-CoDe is not clear in terms of description or experimental results. Most of the results show that C-Code offers a lower running time but is much poorer in quality. The description says adding conditions will help to achieve a lower $N$ value, but the reason is not well described.
6. Figure 2 might be not clearly explained. What does it mean by "lowest divergence with lower N"? Why lower $N$ is important for CoDe since CoDe has the total sampling steps of $N \times B$ not only $N$ like BoN? How to understand the "lowest divergence" in this case?

**Questions:**

The weaknesses already include the questions.

---

### Official Review · Reviewer_GQzW · 2024-11-03

**Soundness:** 3
**Presentation:** 3
**Contribution:** 2
**Rating:** 5
**Confidence:** 5

**Summary:**

The paper presents a new inference-time method for generating images with guidance from a reward function -- i.e., the target distribution is P_pretrained(x) * exp(lambda * r(x)), where lambda is the scale of the KL regularization term in the standard soft RL formulation. Specifically, the paper leverages the fact that reward of x0 predicted from xt (in terms of sampling with diffusion models) is a lower bound of the value function. With this, the authors propose CoDE to do a beam-search-style inference: starting from some xt, it samples a batch of sub-trajectories till time t-tau and pick the one with the greatest value V(x_{t-tau}). In cases when the reward deviates greatly from the pretrained diffusion model, the authors propose a variance dubbed C-CoDe, to denoise the image from a slightly noised version of a reference image, instead of from a Gaussian noise (as in standard diffusion sampling). The authors demonstrate the effectiveness of their method on several tasks, mostly those with a reward that aims to match the generated images with a reference image (using CLIP or some similar feature extractor).

**Strengths:**

The presented method is simple and tuning-free, and it can be used in the common scenario where one wants to generate something similar to a given reference. The reasoning behind this proposed method is also straightforward and clear.

**Weaknesses:**

The paper proposes two algorithms: CoDe that works with generic rewards and C-CoDe that only works with a reference image. I feel that the paper is written in a way that blurs the boundary between these two settings. For instance, the abstract reads "...we explore...C-CoDe, that circumvents the need for differentiable guidance functions and model finetuning", yet the baseline methods it implies, like DDPO, deal with generic settings, not the setting with a reference target. The appropriate way to describe the method, I believe, is to say that **for this specific setting** the authors figure out C-CoDe with advantages blah blah.

The paper does mention many methods for reward finetuning, like DDPO [1] (PPO for diffusion finetuning) and ReFL [2] (direct stochastic reward optimization). ReFL and similarly methods like DRaFT [3] /AlignProp [4] indeed converge at a reasonable speed (probably less than 100 updates). Yet, the paper does not compare their method against these baselines.

Regarding tuning-free methods, I also feel it is nice to cite some from the probabilistic method literature, for instance those with twisted SMC (e.g., [5]), as I found them very similar to the approach here.


[1] Training Diffusion Models with Reinforcement Learning. Kevin Black, Michael Janner, Yilun Du, Ilya Kostrikov, Sergey Levine. ICLR 2024.

[2] ImageReward: Learning and Evaluating Human Preferences for Text-to-Image Generation. Jiazheng Xu, Xiao Liu, Yuchen Wu, Yuxuan Tong, Qinkai Li, Ming Ding, Jie Tang, Yuxiao Dong. NeuIPS 2023.

[3] Directly Fine-Tuning Diffusion Models on Differentiable Rewards. Kevin Clark, Paul Vicol, Kevin Swersky, David J Fleet. ICLR 2024.

[4] Aligning Text-to-Image Diffusion Models with Reward Backpropagation . Mihir Prabhudesai, Anirudh Goyal, Deepak Pathak, Katerina Fragkiadaki . https://arxiv.org/abs/2310.03739.

[5] Practical and Asymptotically Exact Conditional Sampling in Diffusion Models. Luhuan Wu, Brian L. Trippe, Christian A. Naesseth, David M. Blei, John P. Cunningham. NeurIPS 2024.

**Questions:**

Please see weakness section.

---

### Note · Authors · 2024-11-26

**Comment:**

Dear Reviewers and AC,

We are encouraged by the reviewer **Kg9h**’s recognition of the significance of guidance in diffusion models and their appreciation of CoDe’s theoretical groundedness. We appreciate reviewer **GQzW**’s remark on CoDe being a simple and tuning-free approach. Additionally, we are pleased that reviewer **xiW2** highlighted the practicality of our method, noting its broad applicability to diverse downstream tasks without the need for retraining or differentiable guidance functions.

That being said, we would like to clarify our stance on the reviewers’ remarks about the motivation and novelty of our approach. While we do not claim novelty throughout the paper, we are the first to show that combining block-wise optimal sampling with adjustable noise conditioning enhances control over the reward vs. divergence trade-off. Our extensive experimental results demonstrate that our _simple yet effective_ approach offers a balanced trade-off between reward alignment, prompt instruction following, and inference cost, standing on par or outperforming other state-of-the-art baselines.

However, implementing the requested changes of adding more inference-time guidance-based baselines, demonstrating CoDe’s effectiveness on a non-differentiable reward, and clarifying our advantages over fine-tuning based methods require a major revision and would take more time than the rebuttal period would allow. We have made an informed decision to withdraw our paper to take the time needed to fully incorporate the reviewers' insightful feedback and further improve our work to submit it to a future venue.

We would like to express our heartfelt gratitude to the reviewers for their time and thoughtful feedback, which have been instrumental in improving the quality of our work and guiding its future direction.

Kind Regards,

Authors

**Withdrawal Confirmation:**

I have read and agree with the venue's withdrawal policy on behalf of myself and my co-authors.